# Matrix Pencil Method for Vital Sign Detection from Signals Acquired by Microwave Sensors

**DOI:** 10.3390/s21175735

**Published:** 2021-08-26

**Authors:** Somayyeh Chamaani, Alireza Akbarpour, Marko Helbig, Jürgen Sachs

**Affiliations:** 1Time-Domain Electromagnetics Laboratory, Faculty of Electrical Engineering, K.N. Toosi University of Technology, Tehran 1631714191, Iran; a.akbarpour@mail.kntu.ac.ir; 2Biosignal Processing Group, Technische Universität Ilmenau, 98693 Ilmenau, Germany; marko.helbig@tu-ilmenau.de; 3Electronic Measurements and Signal Processing Group, Technische Universität Ilmenau, 98693 Ilmenau, Germany; juergen.sachs@tu-ilmenau.de; 4ILMSENS GmbH, 98693 Ilmenau, Germany

**Keywords:** artery pulsation monitoring, heart rate variability, matrix pencil method, microwave sensor, variational mode decomposition, vital sign signal processing

## Abstract

Microwave sensors have recently been introduced as high-temporal resolution sensors, which could be used in the contactless monitoring of artery pulsation and breathing. However, accurate and efficient signal processing methods are still required. In this paper, the matrix pencil method (MPM), as an efficient method with good frequency resolution, is applied to back-reflected microwave signals to extract vital signs. It is shown that decomposing of the signal to its damping exponentials fulfilled by MPM gives the opportunity to separate signals, e.g., breathing and heartbeat, with high precision. A publicly online dataset (GUARDIAN), obtained by a continuous wave microwave sensor, is applied to evaluate the performance of MPM. Two methods of bandpass filtering (BPF) and variational mode decomposition (VMD) are also implemented. In addition to the GUARDIAN dataset, these methods are also applied to signals acquired by an ultra-wideband (UWB) sensor. It is concluded that when the vital sign is sufficiently strong and pure, all methods, e.g., MPM, VMD, and BPF, are appropriate for vital sign monitoring. However, in noisy cases, MPM has better performance. Therefore, for non-contact microwave vital sign monitoring, which is usually subject to noisy situations, MPM is a powerful method.

## 1. Introduction

In many applications, such as non-obtrusive home care and patient monitoring, sport, military, automotive, security/through the wall and rescue, the remote detection and analysis of vital signs are very important. The most effective modality in remote vital sign monitoring is microwaves. Although recent advances in complementary metal-oxide-semiconductor (CMOS) technology have made microwave technology affordable and publicly available [1], finding a robust signal processing algorithm to extract vital signs from microwave sensors is still an open research area [2].

Among different signal processing methods, fast Fourier transform (FFT) is a very basic method to retrieve respiration and heartbeat rate [3]. However, FFT suffers from poor spectral resolution [4]. Since the heartbeat and harmonics of breathing or intermodulation components are closely spaced, the spectral resolution in heartbeat extraction is very important. Long time windows in FFT lead to good resolution. Contrarily, since the heartbeat and respirations are non-stationary, we are interested in short windows to obtain a time-frequency representation of the signal. A modified version of FFT on the base of time window variation has been proposed to mitigate the poor resolution of FFT due to short time windows [5]. However, it provides the average of heartbeats over 2–5 s windows and not the instantaneous heartrate variability (HRV). High-resolution compressed sensing-based methods were applied to reduce the time window to 5 s [6]. However, they also ignore heart rate variations in the 5 s intervals. Another method to overcome the spectral resolution problem of FFT is chirp Z-transform [4]. However, chirp Z-transform has more computations and is generally slower than FFT. In addition, it was shown in [4] that chirp Z-transform needs a preprocessing filter based on a moving target indicator (MTI) to be efficient. Other techniques have been proposed, including empirical mode decomposition (EMD) [7] and blind source separation techniques such as principal component analysis (PCA) [8], singular value decomposition (SVD) [9] and singular spectrum analysis (SSA) [10] of the signal, and then applying FFT to the decomposed parts to find the vital signals. However, these techniques also need a high computation time. In [11], an adaptive harmonic-comb filter was used for heartbeat estimation. This method applies an optimization procedure and is also time consuming.

Another alternative method for spectral estimation is the matrix pencil method (MPM) [12]. MPM is a well-developed technique proposed in the 1990s [13]. It models the signal as a linear combination of damping exponentials [14]. However, damping behavior is not a necessity, and sinusoidal functions could be considered as a special case of an exponential function with zero damping factor [15].

MPM, which is an efficient version of Prony’s method [16], is a parametric signal model. Against the basic autoregressive (AR) model that is suitable only for stationary signals [17], Prony’s method and MPM can model both stationary and non-stationary signals. The criterion in using MPM is that the signal has to be fitted by a summation of damping exponentials. As long as the signal is fitted, it is not important how stationary it is. However, detrending, segmentation, and filtering may improve the fitting, especially in noisy conditions. Therefore, against “short-time Fourier transform” and “time-variant autoregressive” which are necessary for non-stationary signals, a “short time MPM” is not a necessity for non-stationary signal analysis. However, if one is interested in which natural poles arise at what time, a “short time MPM” must be implemented [18,19].

MPM has been used in several research areas such, electromagnetic scattering [20], angle of arrival estimation [21], radar cross section (RCS) estimation [22], and inverse synthetic aperture radar (ISAR) imaging [23]. Although Prony’s method has been used for electrocardiogram (ECG) signals [17], to the best of our knowledge, MPM has not yet been used for the processing of vital signals.

MPM has been compared with FFT in terms of the accuracy of frequency estimation in [15], and it has been shown that the frequency estimation error of MPM is less than FFT. While FFT resolution is always limited by the length of the considered signal in the time domain, MPM is not characterized by a fixed frequency resolution [24]. The disadvantage of MPM is that it needs a minimum level of SNR. However, usually, this shortage might be mitigated by applying some pre-filtering [14,25]. Therefore, for vital sign analysis, MPM could be considered as a conciliatory alternative, which compromises computation burden and spectral resolution.

In this paper, we applied MPM to the reflected signal from human subjects recorded by microwave sensors. Two types of microwave stimulations are investigated: first, a publicly available database (GUARDIAN) published in 2020 [26], which uses a continuous wave (CW) sensor in 24.17GHz; and second, an ultra-wideband (UWB) sensor (100 MHz–6.5 GHz), which we tested in the laboratory of medical microwave sensing in Ilmenau University of Technology.

After required preprocessing, MPM is applied to the recorded signal, i.e., the signal is estimated by a summation of damped/non-damped exponential components. The only criterion is that the signal must not be very noisy so that it could be fitted by MPM. All of the physiologically meaningful components correspond to different natural poles of signal, e.g., breathing and its harmonics, heartbeat and its harmonics, etc. Since breathing has a higher energy level and usually covers the heartbeat signal, the extraction of the heartbeat signal is usually more challenging. To extract the heartbeat, the breathing-related signal parts have to be identified and removed from the captured data after it got rid of the stationary background. For that purpose, the pole with the highest energy within the frequency range 0.1–0.4 Hz is selected, for which the second and third-order multiples could also be detected among all the natural poles. The interval [0.1, 0.4 Hz] is selected to cover normal adult respiration at rest, which varies between 12 and 20 bpm [27]. However, for non-resting cases of adults, e.g., during exercise, or infants, the upper value should increase to 0.8 Hz [11]. The related frequency *f_b_* of the major pole is assigned to the breathing rate. This pole and all poles with multiples of *f_b_* represent the breathing signal, which is finally subtracted from the measured signal. After removing the breathing pole and its harmonics, the strongest signal in the range of 0.8–2 Hz is considered as the heartbeat (*f_h_*).

To perform a fair comparison with more recent algorithms, we also implemented a variational mode decomposition (VMD)-based method and compared its performance with MPM. VMD is a novel variational method proposed in 2014 [28], and its performance is better than SSA and different versions of EMD [29,30]. VMD has been recently applied for many time-frequency analysis applications, such as rotating machines condition monitoring [31], chatter frequency identification and amplitude tracking [32], power transformer winding assessment [33], third heart sound detection [30], heart rate estimation of wrist-type photoplethysmography (PPG) [29], ECG signal processing [34], apnea detection from ECG [35], and through-wall vital sign tracking [36]. Although VMD is more robust than EMD against mode mixing [37], it still incapable of decomposing the modes that overlap in the Fourier spectrum [38]. Even variational nonlinear chirp mode decomposition (VNCMD), as a newer version of VMD [37], does not solve the shortcoming of VMD in modes with frequency overlap and noisy conditions. The optimum number of decomposition modes and denoising methods are further open issues for VMD [38].

Our results show that, when the heartbeat is strong enough and is not covered with breathing harmonics, MPM and VMD and even conventional bandpass filtering (BPF) have a similar performance. Monitoring of the carotid artery by contact antennas is an example of this case. However, in scenarios where the heartbeat signal is weak or covered by harmonics of breathing or other noises, MPM is superior compared to VMD and BPF methods. We experienced such an example when a sensor was placed at 1m distance from a subject during abdominal crunches, where his third harmonic of breathing was very strong and is very close to the heartbeat. It was observed that MPM was more successful in recovering the heartbeat.

MPM also has less complexity compared to adaptive continuous wavelet [39], which needs much tuning and is too sensitive to SNR. Furthermore, since wavelet base techniques still use widowing (variable-sized regions) [40], they still fail to obtain high resolution. MPM also has less computational burden compared to PCA [41] and bootstrap-based generalized warblet transform [42].

In the rest of this paper, the data acquisition setup is described in Section 2. The signal processing algorithms including preprocessing and post-processing are presented in the Section 3. Section 4 presents the results and discussion, and finally Section 5 concludes the paper.

## 2. Data Acquisition Setup

### 2.1. CW Sensor

The CW sensor data were taken from the GUARDIAN dataset, where a CW sensor at 24.17 GHz using a six-port network captured the signals [26]. The sampling rate of this dataset is 2000 Hz. In addition to the microwave sensor, the GUARDIAN project setup also has an ECG sensor, a respiration sensor, and a digital stethoscope. Eleven test subjects with different positions of microwave sensor beam and digital stethoscope placement, e.g., the carotid, the back, and several frontal positions on the thorax, were studied. In all tests, the sensor had a 20 cm distance from the body. A detailed description of this setup and the resulted dataset has been reported in [26].

### 2.2. Ultra Wideband Sensor

The UWB setup used in the measurements of this paper was described completely in [43]. The M-sequence UWB sensor used here is a baseband system (100 MHz–6.5 GHz bandwidth) with one transmitter and two receivers (Ilmsens GmbH, Ilmenau, Germany), whereas only one receiver was used in our measurements. This sensor produces pseudo-noise stimulation signals (M-sequence) using a high-speed shift register. This kind of UWB signal is very robust against noise and jitter and therefore is very proper for weak movement detection [44]. In addition, compared to the CW sensor, the high bandwidth of the UWB sensor provides fine range resolution and can be used in localization applications. The output power of the sensor is +3 dBm, and the antennas are passive elliptical dipole antennas (16 mm × 40 mm). These antennas were differentially fed using a passive balun. To compare the sensor results with ECG, a synchronization unit was designed and implemented. This unit included a small and powerful Linux system ODROID U3 (Hardkernel Co. Ltd., AnYang, Korea) with 1.7 GHz ARM Qual-Core processor and an analog to digital convertor (ADC) board based on the ADS1198 (Texas Instruments, Dallas, TE, USA, 8 channels, 16 bit). Before ADC, the analog ECG signal was amplified using a commercial system g.BSamp (g.tec medical engineering GmbH, Schiedlberg, Austria). The UWB measurement system is shown in Figure 1.

## 3. Materials and Methods

In every instance of microwave-based vital sign monitoring, a general flowchart should be followed (Figure 2). The type of sensor only affects the preprocessing stage. After preprocessing, signal processing is applied. Finally, post-processing is necessary to extract the vital sign from the processed data, e.g., which poles in MPM or which intrinsic mode functions in VMD should be kept or discarded, and how to calculate the heartbeat.

All parts of this flowchart will be explained in this section, and some illustrative examples will be given in Section 4.

### 3.1. Continious Wave Sensor Signal Preprocessing

The CW sensor used in the GUARDIAN dataset receives in-phase (I) and quadrature (Q) components as follows:(1)$$I=A_{I}\cos(\frac{4\pi r(t)}{\lambda }+\Phi_{I})+B_{I}\;\;\;$$
(2)$$Q=A_{Q}\sin(\frac{4\pi r(t)}{\lambda }+\Phi_{Q})+B_{Q}\;\;\;$$
where $B_{I},B_{Q}$ represent DC offset, r(t) represents the time-varying displacement of the target, and λ is the wavelength in free space. These formulas are exact for a single point target and in the case of multipath propagation, the modulation term r(t) is a complicated mixture from all targets within the radar beam. However, for a first-order estimation, we considered the formula for the single-point target. The corresponding preprocessing is shown in Algorithm 1. The first step is ellipse fitting which compensates for the unbalance between the I and Q ports of the sensor. These unbalance result from hardware imperfections [45]. According to the ellipse fitting method introduced in [45], an ellipse can be fitted to I and Q components. In an ideal case (without any unbalance), this ellipse must be a circle centered at the origin, while in practical erroneous cases, the data are fitted to an off-centered and rotated ellipse. When the data are fitted to the corresponding ellipse, first a transform is performed to move the data to the center and compensate for the DC-offset. In the second step, the ellipse is rotated to compensate for the phase imbalance. Finally, the gain imbalance is compensated for, and a circle centered at the origin will be produced. Then, Arctangent demodulation is applied to achieve r(t).

In Figure 3, the results of ellipse fitting for different data sizes of person 10 ID = 10–40–44 from the GUARDIAN database are plotted. As observed, the ellipses do not show very strong variations, especially for longer windows, e.g., 15 and 43 s. However, if somebody is interested in instantaneous unbalance compensation, only 5 points are required for ellipse fitting.
**Algorithm 1.** CW Sensor Signal Preprocessing1.Ellipse fittingDC offset compensationPhase imbalance compensationAmplitude imbalance compensation2.Arctangent demodulation

### 3.2. Ultra Wideband Sensor Signal Preprocessing

The preprocessing step of the UWB sensor signal is shown in Algorithm 2. In the UWB sensor, the output of data acquisition is a matrix called Raradgram or fast time-slow time matrix (R). Each column of Radargram is an impulse response. These impulse responses are recorded sequentially. Accordingly, Radargram is a P × N matrix where P is the number of range bins in each impulse response (the signal received by the sensor) and N is the number of impulse responses. Therefore, the time step between two successive samples in one column is in the order of “sub-nanosecond” (fast time/propagation time) and the time step between two successive samples in one row is in the order of “sub-second” (slow time/observation time). A Radargram recorded by the sensor placed on the carotid artery is shown in Figure 4. The sampling frequency over observation time/slow time is 50 Hz and the observation time step is 0.02 s. The propagation time step is $\frac{1}{f_{c}}$ where $f_{c}=$ 13 GHz in our device. The artery motion is monitored throughout the 50 s (i.e., N = 2500). In the UWB measurement scenario, the Tx and Rx antennas are closely spaced together with similar polarizations (i.e., both in parallel with the artery). In this application, for the sake of simplicity, time zero is estimated from crosstalk between the antennas of known distance. In the case of array processing and microwave imaging, it has to be considered with more care. Correct time zero setting requires precise consideration of the antenna’s radiation center, which is beyond the scope of this paper.
**Algorithm 2.** UWB Sensor Signal Preprocessing
      1. Building the Radargram
      2. Static clutter removal
      3. Target range bin determination
      4. Getting the reflected signal of target over the observation time


In Figure 4a, the dark blue line is the result of antenna cross talk. This coupling and other unwanted signals, which are stationary during the observation time, e.g., reflection from stationary parts of the body and reflection from static obstacles in the surrounding environment, are removed in the “static clutter removal” step. For static clutter removal, in each row of the Radargram, the average of that row (range bin) over the observation time is supposed as background. Therefore, the static background will be removed according to the following equation:(3)$$R(m,n)=R(m,n)-\frac{1}{N}\sum_{n=1}^{N}R(m,n)$$
where *R*(*m*,*n*) represents the Radargram matrix, “*m*” refers to the range bin number, and “*n*” refers to the observation number. The Radargram for the carotid artery after clutter removal is shown in Figure 4b. In this test with the UWB sensor, antennas are placed directly above the naked skin. Since the carotid is very close to the skin and also the antennas, the fluctuations of the carotid are almost in the same range bin as the antenna coupling effect (about 1 ns), but with a much smaller amplitude. It should be noted that due to the long ringing of the antenna impulse response, we cannot obtain the exact depth of the artery in this Radargram.

**Figure 4 sensors-21-05735-f004:**
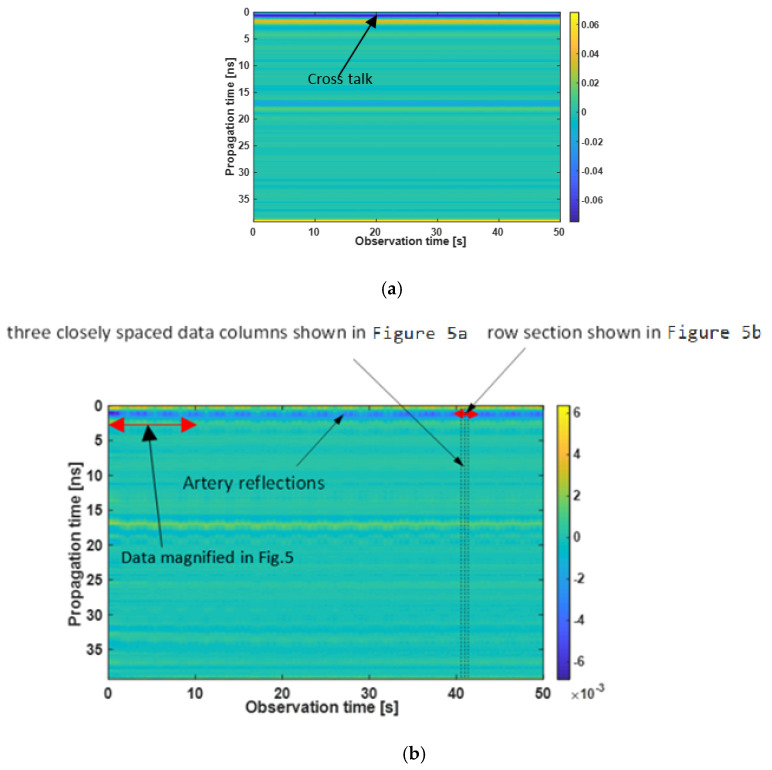
Radargram (**a**) raw data and (**b**) data with static background removed (note the different color scaling of both Radargram).

As can be seen in Figure 4b, after static clutter removal, the fluctuations of the received signal, which is the result of breathing and heartbeat, in some range bins (between 0 and 4 ns) are visible. For a single-target scenario, this duration is the time width of the received signal, which is mainly determined by the antenna pulse response in our case. For better understanding, one typical received signal (one column of Radargram) is shown in Figure 4b and Figure 5. Indeed, all of the points in the time width of the received signal (between 1 and 3 ns) have fluctuations due to vital signs. There are different ways to decide about the proper range of the target [44,46,47]. In this paper, the row with the maximum standard deviation is selected as the target range bin. As observed in Figure 5a, weak movements of the artery lead to small variations in the time of arrival and amplitude of the signal. If we look at the row corresponding to the target in different observation times, these weak movements resemble modulations created by weak movements such as the heartbeat (Figure 5b).

After preprocessing and obtaining the signal, including micro movement information relating to the body, e.g., breathing, artery pulsation, unwanted non-stationary clutter, and noise, the proper signal processing method must be applied to extract the wanted vital signs. The signal processing method is general and does not depend on the sensor type. The main signal processing method applied in this paper is MPM. Additionally, to make a comparison with a recently proposed method, VMD is applied. In the following, MPM is explained thoroughly. A short description of VMD and related references is also provided.

### 3.3. Matrix Pencil Method

The matrix pencil method is a method for the decomposition of noisy signals to a sum of damping exponentials. The observed signal y(T) is written as follows [13]:(4)$$y(T)=x(T)+n(T)\approx \sum_{i=1}^{M}R_{i}\exp(s_{i}T)+n(T),\;\;0\leq T\leq T_{\mathrm{end}}\;\;\;\;$$
where T is the observation time, n(T) is the noise, x(T) is the signal, and $R_{i}$ represents the residues or complex amplitudes. Each exponential $R_{i}\exp(s_{i}T)$ is called a pole. The $s_{i}$ is defined as follows:(5)$$s_{i}=\alpha_{i}+j\omega_{i}$$
where $\alpha_{i}$ and $\omega_{i}=2\pi f_{i}$ represent the damping factor and angular frequencies, respectively. Since y(T) is a real-valued signal, $R_{i}$ and $s_{i}$ must appear in complex conjugate pairs. For a sampled signal with a sampling interval of $T_{s}$, the observed signal can be written as
(6)$$\begin{array}{l} y(kT_{s})=x({\mathrm{kT}}_{s})+n({\mathrm{kT}}_{s})\approx \sum_{i=1}^{M}R_{i}z_{i}^{k}+n({\mathrm{kT}}_{s}),\mathrm{and}\\ k=0,1,\ldots ,N-1 \end{array}$$
(7)$$z_{i}=e^{s_{i}T_{s}}=e^{(-\alpha_{i}+j\omega_{i})T_{s}},\;\;\;\;i=1,2,\ldots ,M$$

To implement the MPM, a Hankel matrix is constructed from noisy data as follows:(8)$$\left[Y\right]={\left[\begin{array}{l} y(0)\;\;\;\;\;\;\;\;\;\;\;\;\;\;\;\;\;\;\;\;\;\;\;\;\;\;y(1)\;\;\;\;\;\;\;\;\;\;\;\;\;\;\;\;\ldots \;\;\;\;\;\;\;\;\;\;\;\;\;\;\;y(L)\\ y(1)\;\;\;\;\;\;\;\;\;\;\;\;\;\;\;\;\;\;\;\;\;\;\;\;\;\;y(2)\;\;\;\;\;\;\;\;\;\;\;\;\;\;\;\;\ldots \;\;\;\;\;\;\;\;\;\;\;\;\;\;\;y(L+1)\\ \;\;\;\;\;\vdots \;\;\;\;\;\;\;\;\;\;\;\;\;\;\;\;\;\;\;\;\;\;\;\;\;\;\;\;\;\;\vdots \;\;\;\;\;\;\;\;\;\;\;\;\;\;\;\;\;\;\;\;\;\vdots \;\;\;\;\;\;\;\;\;\;\;\;\;\;\;\;\;\;\;\;\;\;\;\vdots \\ y(N-L-1)\;\;\;\;\;y(N-L)\;\;\;\;\;\;\;\;\;\;\;\;\ldots \;\;\;\;\;\;\;\;\;\;\;\;\;\;y(N-1) \end{array}\right]}_{{}_{(N-L)\times L}}$$

This Hankel matrix is a combination of two matrixes of $Y_{1}$ and $Y_{2}$, which are obtained by deleting the last and first columns of the *Y* matrix, respectively. L represents the pencil parameter. Generally, M<L<N−M (*M* is the number of damping exponentials that fit the observed signal). However, for efficient noise filtering, N3<L<N2 [48]. In the next step, an SVD of the matrix Y is performed as follows:(9)$$Y=U\sum V^{H}$$

In (9), the superscript “*H*” denotes the conjugate transpose, and *U* and *V* are unitary matrices including eigenvectors of $YY^{H}$ and $Y^{H}Y$, respectively. ∑ is a rectangular diagonal matrix composed of the singular values ($\sigma_{i}$) of Y. These singular values construct the diagonal values of ∑ in descending order ($\sigma_{1}\geq \sigma_{2}\geq \cdots \geq \sigma_{\min}$). Usually, singular values beyond *M* are set to zero. Now, one should decide about *M*. For this purpose, consider singular value $\sigma_{M}$ as follows:(10)$$\frac{\sigma_{M}}{\sigma_{1}}\approx tol$$
where tol shows a threshold level for considering singular values below that as noise and discard them. Although the thermal noise level could be a small value, in the absence of any other noise, we can consider the analog to digital converter quantization error as the noise level. Therefore, for a q-bit ADC with ±1 V range, a rough estimation of tol is $\frac{1}{2^{q}}$. For an ADC with 24bit, it will result in $tol=\frac{1}{2^{24}}$. However, the reported values in the GUARDIAN dataset are accurate up to 4 significant digits, which means there are extra noises, in addition to quantization noise. Therefore, *tol* will be $\frac{1}{10^{4}}$ [14]. For the UWB sensor, however, its internal correlation operation affects the noise level. In the case of the UWB sensor, it is easy to determine the noise level by looking for the variance at signal segments, which are not affected by moving targets. This value for our sensor was also 0.0001.

Thus, singular values with the values of (10) below tol correspond to noise and should not be used to reconstruct the signal. Discarding the singular values beyond $\sigma_{M}$ acts as a built-in filter. The filtered matrix $V_{s}$ contains only *M* dominant right singular-vector of *V*:(11)$$V_{s}=[v_{1},v_{2},\ldots v_{M}]$$

The remaining singular vectors from M+1 to L are discarded.
(12)$$Y_{1}=U\sum_{s}V_{1s}^{H}\;\;\;\;\mathrm{and}\;\;\;\;Y_{2}=U\sum_{s}V_{2s}^{H}$$
where the superscript H stands for Hermitian conjugate. Matrices $V_{1s}$ and $V_{2s}$ are built by deleting the last and first rows of $V_{s}$, respectively. $\sum_{s}$ is a diagonal matrix consisting of only *M* dominant values of ∑. The poles $z_{i}$ introduced in (5) are generalized eigenvalues $\lambda_{i}$ of the following matrix equation:(13)$${\left\{Y_{2}-\lambda Y_{1}\right\}}_{L\times M}=0\;\;\;\mathrm{or}\;\;\;\;Y_{1}^{+}Y_{2}=\lambda I_{M\times M}$$

In (10), $Y_{1}^{+}$ is the Moore–Penrose pseudoinverse of $Y_{1}$. Frequencies and damping factors of exponentials are calculated from eigenvalues as follows:(14)fi=Im{λi}2πTs and αi=Re{λi}Ts

After finding *M* and $z_{i}$, the residues $R_{i}$ may be found by polynomial fitting based on a least-squares problem with the Vandermonde matrix of the poles:(15)$$\begin{array}{l} \left[\begin{array}{l} y_{0}\\ y_{1}\\ \vdots \\ y_{N-1} \end{array}\right]=\left[\begin{array}{l} 1\;\;\;\;\;\;\;\;\;\;\;\;\;\;\;1\;\;\;\;\;\;\;\ldots \;\;\;\;\;\;\;\;1\;\;\;\;\;\;\;\;\;\;\;\\ z_{1}\;\;\;\;\;\;\;\;\;\;\;z_{2}\;\;\;\;\;\ldots \;\;\;\;\;\;\;z_{M}\\ \vdots \;\;\;\;\;\;\;\;\;\;\;\;\;\;\;\;\vdots \;\;\;\;\;\;\;\;\;\ddots \;\;\;\;\;\;\vdots \\ z_{1}^{N-1}\;\;\;\;\;\;z_{2}^{N-1}\;\;\;\ldots \;\;\;\;z_{M}^{N-1} \end{array}\right]\left[\begin{array}{l} R_{1}\\ R_{2}\\ \vdots \\ R_{M} \end{array}\right]\\ Y=Z\;R\\ R={\left(Z^{H}Z\right)}^{-1}Z^{H}Y \end{array}$$

After calculating the natural frequencies and residues of the signal using MPM, in a post-processing step, we could decide about the desired frequencies and discard the rest. For artery pulsation applications, the desired frequencies are heartbeat and its harmonics. First, breathing frequency is estimated (the signal corresponding to the highest amplitude-if its first or second harmonics are also present is considered as breathing). After removing the breathing and its harmonics, in the resting cases, the signal with the highest amplitude in [0.8–2 Hz] is considered as the heartbeat.

After choosing the heartbeat and its harmonics and discarding other natural poles, the pure artery pulsation can be reconstructed. The more harmonics you can detect, the more accurately the pulse can be reconstructed. With a sampling frequency of 50 Hz, a maximum frequency of 25 Hz is achievable. However, usually, only the first four harmonics have enough SNR to be detected.

### 3.4. Variational Mode Decomposition Method

In VMD [28], the signal g(t) is considered as a sum of K intrinsic narrowband intrinsic mode functions (IMF) as follows:(16)$$g(t)=\sum_{k=1}^{K}u_{k}(t)$$

Intrinsic mode functions are amplitude-modulated-frequency-modulated (AM-FM) signals:(17)$$u_{k}(t)=A_{k}(t)\cos(\varphi_{k}(t))$$
where the phase $\varphi_{k}(t)$ is a non-decreasing function and therefore the instantaneous frequency $f_{k}(t)=\frac{1}{2\pi }\frac{\partial \varphi_{k}(t)}{\partial t}$ is non-negative. The envelope $A_{k}(t)$ is also non-negative and both the envelope and the instantaneous frequency vary much slower than the phase $\varphi_{k}(t)$ [28].

We used the “vmd” function in MATLAB 2020 to compute the IMFs of the signal. After preliminary examinations, we chose K = 9. After decomposing of signal to IMFs, Hilbert–Huang transform was applied on IMFs to perform spectral analysis. After calculating the spectral contents of each IMF, in the post-processing stage, the mean values of instantaneous frequency and instantaneous energy of each IMF were calculated. Similar to post-processing of MPM, the IMF with the highest energy in [0.1–0.4 Hz]—for normal physical activities (i.e., not exercising)—was considered as breathing, and its harmonics are discarded. Among the remaining IMFs, the strongest IMF with mean of instantaneous frequency in [0.8–2 Hz] was selected as the IMF that corresponds to the heartbeat. The mean instantaneous frequency of this IMF was considered as “average heartbeat”. Then, the IMFs whose mean frequency was close to multiples of “average heartbeat” were considered as harmonic IMFs. We search for the first four harmonics of the heartbeat. Finally, the summation of these IMFs was considered the artery-pulsation wave.

## 4. Results and Discussion

### 4.1. CW Sensor Results

In this subsection, we focus on the application of MPM and VMD on the GUARDIAN project dataset. GUARDIAN dataset has 11 subjects, and each subject has been monitored in different scenarios where overall, 223 min of data were acquired. For the sake of brevity, we apply our algorithms to only one scenario of each subject.

For better illustration, details of vital sign signals extracted by different algorithms for one scenario of person 10, ID = 9-53-17, are shown in Figure 6, Figure 7, Figure 8 and Figure 9. In this scenario, which belongs to the “after sport” condition, the heartbeat starts from 120 bpm (2 Hz) and decreases to 110 bpm (1.83 Hz) in the first 16s. The breathing rate also decreases and starts from 0.3 Hz. The natural poles of the signal obtained by MPM and the absolute value of Fourier transform of signal in different signal processing stages are shown in Figure 6 and Figure 7, respectively. After detection of the natural poles of signal, the poles related to the breathing and its harmonics must be removed from the signal. After removing the breathing harmonics, a bandpass filter is applied, i.e., the natural poles with frequencies in [0.8–10 Hz] are kept and the rest of the poles are discarded (Figure 6a and Figure 7a). Finally, among the remaining poles, the pole with the highest amplitude in [0.8–2 Hz] is considered as the heartbeat. After detecting the heartbeat, its harmonics are also selected to reconstruct the artery pulsation signal (Figure 6c and Figure 7b). The breathing corresponds to the natural pole with the highest amplitude in the frequency range of [0.1–0.4 Hz]. According to Figure 7a, the breathing frequency is 0.3 Hz. Figure 7a also shows that even the fourth harmonic of breathing has a higher amplitude than the heartbeat. In addition, the frequencies of the third and fourth harmonic of breathing are about 0.9 and 1.2 Hz, respectively (Figure 6b). Therefore, in such cases, the conventional method of BPF fails, because it considers the strongest signal in [0.8–2 Hz] as a heartbeat, which in this case will be the third harmonic of breathing (0.9 Hz). It is worth noting that removing the breathing and its harmonics is not always possible with FFT-based methods. The reason for this is that in FFT-based methods, with short window sizes, the resolution will be limited, and if one wants to remove the sixth harmonic of breathing (6 × 0.3 = 1.8 Hz), the heartbeat will also be removed.

The results of VMD are also shown in Figure 8. Figure 8a shows the first nine IMFs of signal and Figure 8b shows the Hilbert–Huang transform of these IMFs. As seen in Figure 8b, IMF9 corresponds to respiration. IMF1-IMF3 are high-frequency components without any significant frequency content in [0–10 Hz]. The first step is to find the heartbeat. Again, if we look for the highest energy in [0.8–2 Hz], we will obtain IMF8 as the false heartbeat. One solution is to check that whether the harmonics of the detected heartbeat are also present or not. In this case, the third harmonic of IMF8 is not present, while the third harmonic of IMF7 is present. Therefore, IMF 7 is the heartbeat. As observed from Figure 8a, IMF7 cannot reconstruct the artery pulsation and therefore, its harmonics are also necessary. IMF4-IMF6 are harmonics of the heartbeat. Compared to MPM, VMD gives less precise information about the mode. For example, in MPM, we find three distinct pole sets (with different frequencies) as harmonics of breathing (Figure 6b). However, we only find one IMF (IMF8) as the harmonics related to breathing.

One important parameter in MPM is the proper window (segment) size. Large window sizes may produce an error. The source of error in long windows is the uncertainties in the noisy signal fitting calculations which occur in all algorithms, i.e., VMD, MPM, and BPF. In low noise conditions, a large window size reduces the uncertainties in signal modeling. However, for highly noisy cases, the window size should be smaller. This could be observed in Figure 9a, wherein the time interval [3 s, 4 s] and [10 s, 12 s] the signal level is low. Additionally, if we reduce the window size to 12.5 s, the result will be better (see Figure 9b). However, for other points that have adequate signal amplitude, the large window sizes such as 55 s are still proper. This can be explained for MPM as follows: according to (3), each mode in MPM is an exponential (usually decaying) signal; when the window is too long, the detected pole goes below the noise level; this will add unwanted noise and reduce the accuracy of MPM [49]. Moreover, small window sizes need a short computation time and make real-time heartbeat extraction more feasible. To be conservative, we chose the minimum window size as five times the mode cycle with the lowest frequency [49]. For example, if we aim to extract the breathing, its usual duration is about 2.5–4 s. Therefore, a window size of [12.5–20 s] is appropriate. As can be seen in Figure 9, the breathing signal detected by MPM (Microwave_Respiration) is detectable in both window sizes of 55 s (Figure 9a) and 12 s (Figure 9b). The reason for the good result for breathing in a long window size of 55s is that, compared to the heartbeat, the breathing signal is satisfactorily strong and is not covered by noise.

Another worth point mentioning is that we used the peak-to-peak time difference of successive pulsations to calculate the heartbeat. This method is very sensitive to the shape of pulsation. The shape of pulsation varies based on many conditions, such as thorax effects, venous pulse effect, and signal processing method [50]. In [50], five different templates were defined for pulsation and an advanced template matching algorithm was used to detect HRV. In [51], six templates were introduced and a feature-based correlation algorithm was used to detect HRV. Applying these complicated algorithms for heartbeat calculation increases the accuracy of HRV. However, they are outside of the scope of this paper. Our target is to compare MPM with the other two algorithms of VMD and BPF at the same condition and the same methods of HRV calculations. Figure 10 shows the beat-to-beat HRV obtained using different methods. The data used here belong to the “after sport” scenario which shows a decaying heartbeat. As observed, MPM follows the ECG better than VMD and BPF, and the worst method is BPF. Statistical plots of these data are also plotted in Figure 11. The results show that the MPM heartbeats have the highest correlation (r = 0.89) with the heartbeats obtained by ECG. To gain more insight into the performance of different methods, Bland–Altman analysis between each method and ECG was performed. This analysis was performed using Medcalc software [52]. The Bland–Altman analysis plots the differences between two methods’ results and a reference (here, mean of two methods’ results). The horizontal blue line shows the mean of differences, which for MPM is the lowest value. This means that MPM estimates the heartbeat with the lowest bias. In addition, the MPM results are placed more centrally in comparison with other methods, i.e., heartbeats extracted by MPM are more consistent with ECG and have fewer errors. The root mean square errors (RMSE) of every method with respect to ECG for 11 different scenarios are listed in Table 1. It is obvious that the RMSE of the MPM method is lower than that of VMD and BPF.

**Table 1 sensors-21-05735-t001:** Root mean square error (RMSE) of beat-to-beat heartbeat calculated for different subjects and scenarios from the GUARDIAN project. The related ID that shows the scenario is written in each column. All RMSE values are given in beat/min.

Method	Person 1 ID = 13-55-07	Person 2 ID = 10-18-52	Person 3 ID = 19-12-55	Person 4 ID = 10-41-20	Person 5 ID = 11-51-19	Person 6 ID = 9-10-20	Person 7 ID = 9-4-15	Person 8 ID = 10-6-46	Person 9 ID = 9-20-17	Person 10 ID = 9-40-56	Person 11 ID = 10-40-44	Average
MPM	4.99	6.25	3.39	3.5	1.84	9.81	5.57	3.2	5.68	3.66	4.06	4.72
VMD	14.43	10.54	5.08	9.11	14.47	18.83	7.36	7.4	14.29	4.44	12.46	10.76
BPF	17.64	10.96	14.27	10.63	10.06	11.86	5.8	10.7	11.33	6.97	15.79	11.45

**Figure 8 sensors-21-05735-f008:**
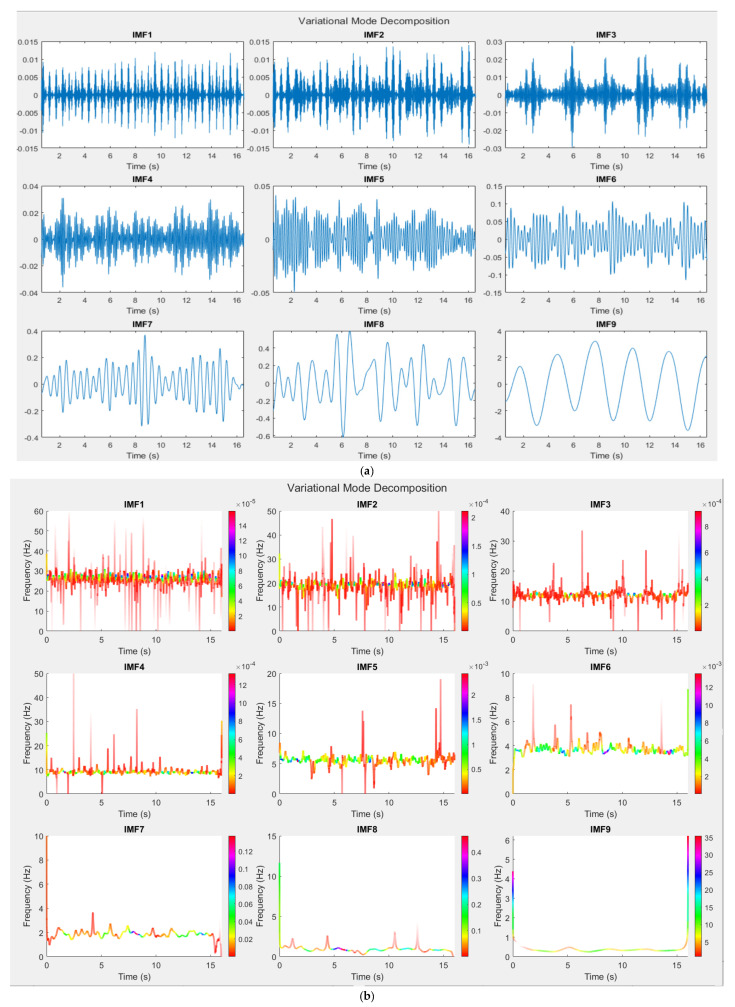
Results of application of VMD on the first 16s of signal captured by CW sensor, GUARDIAN database, person 10, ID = 9-53-17; (**a**) first nine IMFs in time domain and (**b**) Hilbert–Huang transform of first nine IMFs.

**Figure 9 sensors-21-05735-f009:**
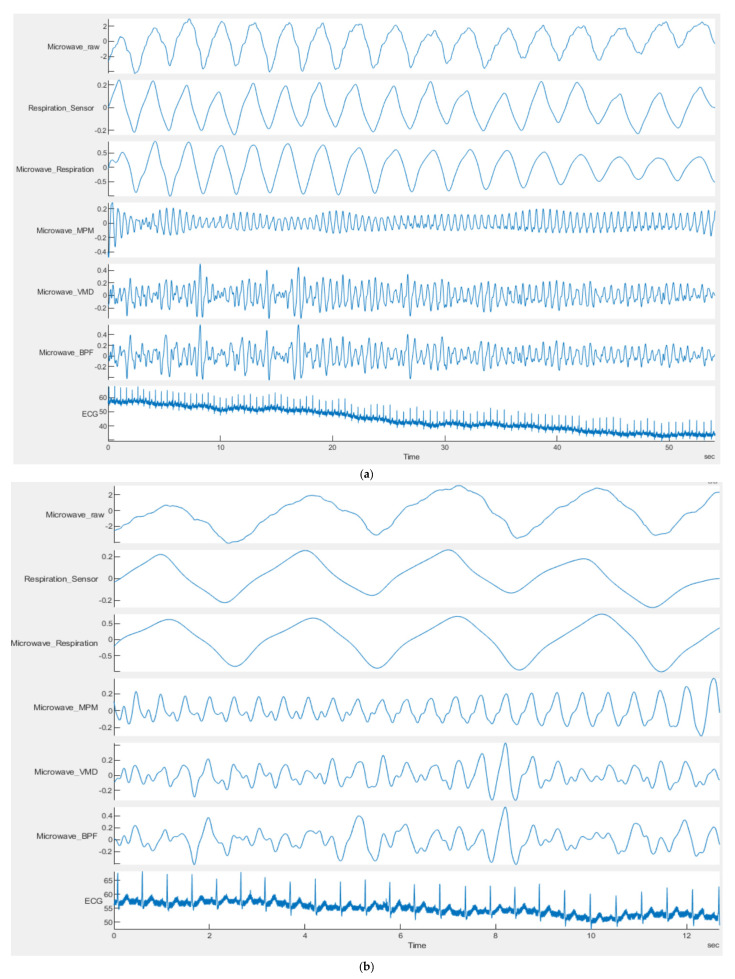
Exemplary of different window length for data of GUARDIAN database (acquired by CW sensor), person 10-ID = 9-53-17; (**a**) 55 s and (**b**) 12.5 s. In both figures, the first row shows the raw data from the microwave sensor. The second row shows the output signal of the respiration senor. The third row shows the respiration signal obtained from MPM. Rows 4–6 show the heart pulsation signal obtained by MPM, VMD and BPF, respectively.

**Figure 10 sensors-21-05735-f010:**
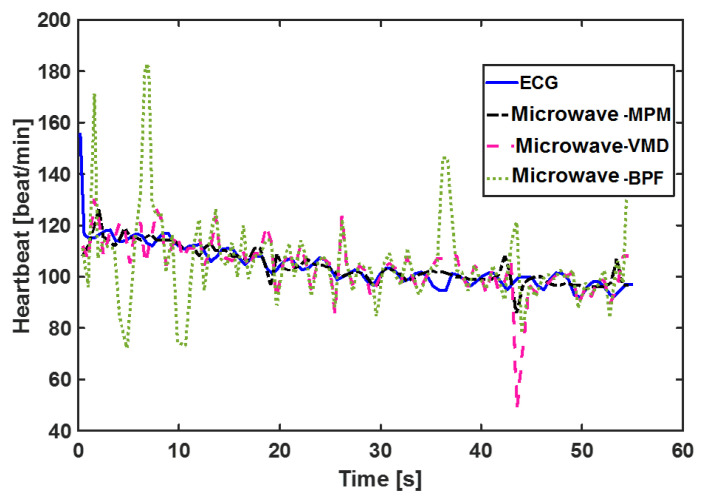
Calculated beat-to-beat heartbeat for person10, ID = 9-53-17 with different signal processing methods. ECG is the gold standard.

**Figure 11 sensors-21-05735-f011:**
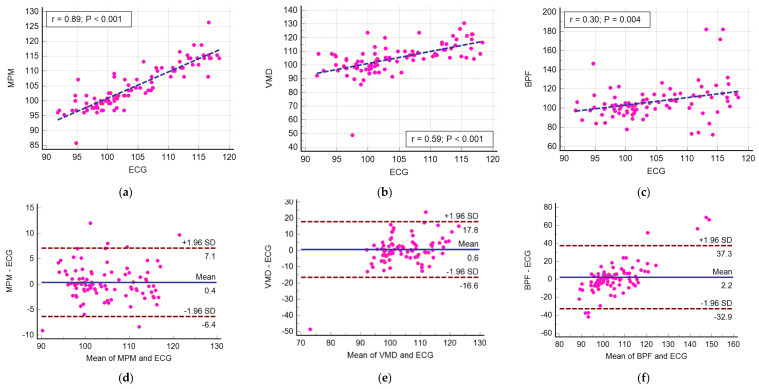
Performance comparison of different signal processing methods for the beat-to-beat heartbeat. Scatter diagram (**a**–**c**), Bland–Altman plot (**d**–**f**). All values are in beats per minute.

### 4.2. Ultra Wideband Sensor Results

In the scenario used for the UWB sensor, two measurement campaigns were performed. In the first measurement campaign, two printed elliptical antennas are placed on the carotid artery of a healthy male subject, 30 years old and 169 cm in height. A carotid artery is a large artery with strong pulsation, usually located very close to the skin, i.e., the sounding signal bears low loss due to thin body tissue. Therefore, in this scenario, the signal is stronger compared to the scenarios the sensor is placed far from the body. Moreover, since the antennas are placed directly on the neck, they receive less effect from breathing, i.e., the chest is not completely in their field of view. Therefore, the signal to noise/clutter ratio is higher than in noncontact scenarios. The resulting pulsations for the UWB sensor are plotted in Figure 12. The very strong and pure heartbeat in this scenario has two important effects: first, segmentation is not necessary and a window size of 50 s does not provide any meaningful difference from a window size of 20 s or less. Second, all methods are successful in reconstructing the artery pulsation and there is no significant difference between different methods. The HRV obtained by different methods is shown in Figure 13, and the errors for this scenario are listed in Table 2 which shows almost similar values.

**Table 2 sensors-21-05735-t002:** Root mean square error of beat-to-beat heartbeat calculated for the carotid artery illuminated by the UWB sensor.

Method	RMSE (beat/min)
MPM	2.59
VMD	2.57
BPF	3.44

In the second measurement campaign of the UWB sensor, a more complicated case was examined, in which the same subject in campaign 1 performed abdominal crunches over a sofa, and two horn antennas were placed 1 m away from the subject, with line of sight, both on the same side of the target and 1 m away from each other and at the same height as the sofa (see Figure 14a). The Radargram is plotted in Figure 14b. As observed, since the subject is non-stationary, a constant range bin cannot be assigned as the range bin of the subject. Therefore, a motion compensation method must be applied to the Radargram before any range bin determination. A conventional method for motion compensation is cross-correlation [53]. This method has been applied for UWB sensing of breathing signals [54]. To implement this motion compensation algorithm, first, a column of the Radargram matrix (one of the received waveforms) was selected as the reference. Theoretically, this reference could be any column. We selected the first column as the reference. Then, the cross-correlation between each column of the Radargram matrix and the reference was calculated. The location of the maximum of the calculated cross-correlation function determines the time shift, which is required to align each column of the Radargram with the reference column. Therefore, each column is circularly shifted according to its required time shift and all columns become aligned and the motion of the subject is compensated. The Radargram after motion compensation is plotted in Figure 14c. Now, the motion-compensated Radargram can be treated as a normal Radargram and its vital sign signals can be extracted. Figure 15 shows the spectrum of the captured signal. As expected, during the exercise (abdominal crunch), both breathing rate and heartbeat are higher than normal, whereas breathing is about 0.6 Hz and its harmonics are strongly present and dominate the heartbeat—this situation is similar to Figure 7. Particularly, the third harmonic of breathing is very close to the heartbeat (~1.6 Hz). As observed, these two components (the third harmonic of breathing and the heartbeat are not distinguishable in conventional Fourier-based vital sign detection algorithms. However, in MPM, since we can easily remove the natural poles related to the harmonics of breathing (red curve in Figure 15), the third harmonic will not be mistakenly considered as the heartbeat. In Figure 15, we select the highest frequency in [1–2.2 Hz] as the heartbeat. We shifted the possible interval of the heartbeat from [0.8–2] to [1–2.2] due to exercising. Figure 16 shows the heartbeat extracted from the UWB sensor by BPF, MPM, and VMD methods. Due to some limitations of our ECG sensor, in this test, a PPG sensor was connected to the finger of the subject during the exercise and used here as the reference signal. The PPG sensor was synchronized with the UWB sensor using an Arduino board. Similarly to dataset of GUARDIAN, the antennas in this experiment were placed far from the subject and the signal to noise ratio decreased (in comparison with the carotid artery experiment in the first campaign). As observed, the MPM is more successful in tracking the heartbeat. For better illustration, the calculated beat-to-beat heartbeat is shown in Figure 17. Since the BPF is not successful at all, its results are not shown. As seen in Figure 17, the MPM method reveals the beat-to-beat heart rate more precisely.

**Figure 12 sensors-21-05735-f012:**
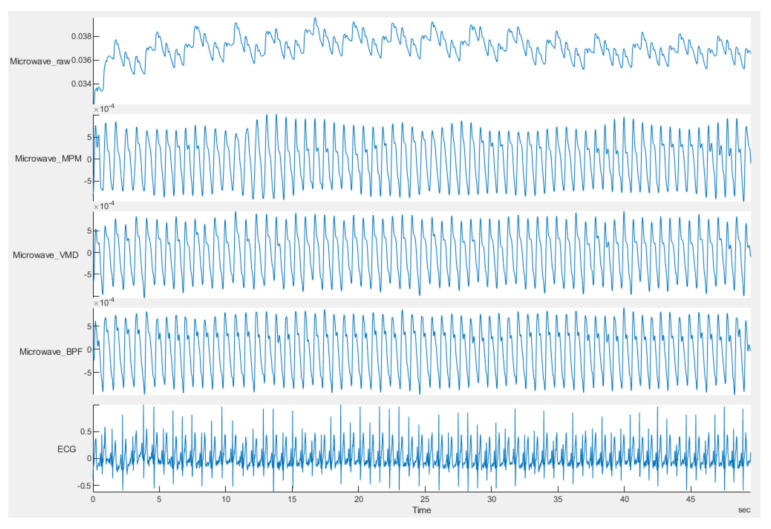
Comparison of carotid signal achieved by the UWB microwave sensor with different methods. The first row shows the raw signal. Rows 2–4 show the result of applying MPM, VMD and BPF on the raw data, respectively. The ECG signal is the gold standard.

**Figure 13 sensors-21-05735-f013:**
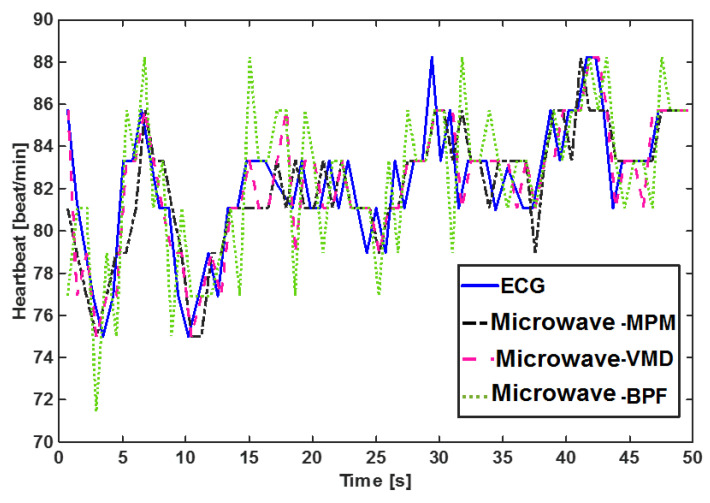
Calculated beat-to-beat heartbeat for the carotid artery by different signal processing methods. The signal was obtained by the UWB sensor.

**Figure 14 sensors-21-05735-f014:**
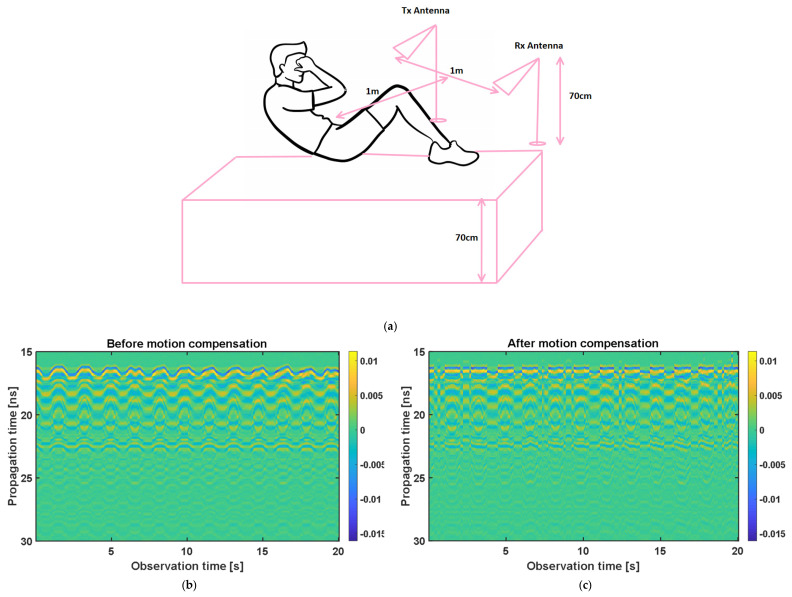
(**a**) Schematic of abdominal crunch measurement scenario. Radargram (**b**) after static background removal but before motion compensation and (**c**) after static background removal and motion compensation.

**Figure 15 sensors-21-05735-f015:**
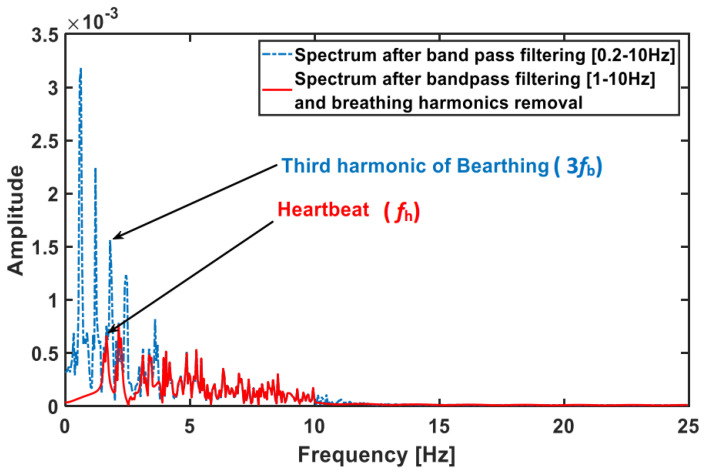
Absolute value of Fourier transform of the signal captured by the UWB sensor during abdominal crunch exercise.

**Figure 16 sensors-21-05735-f016:**
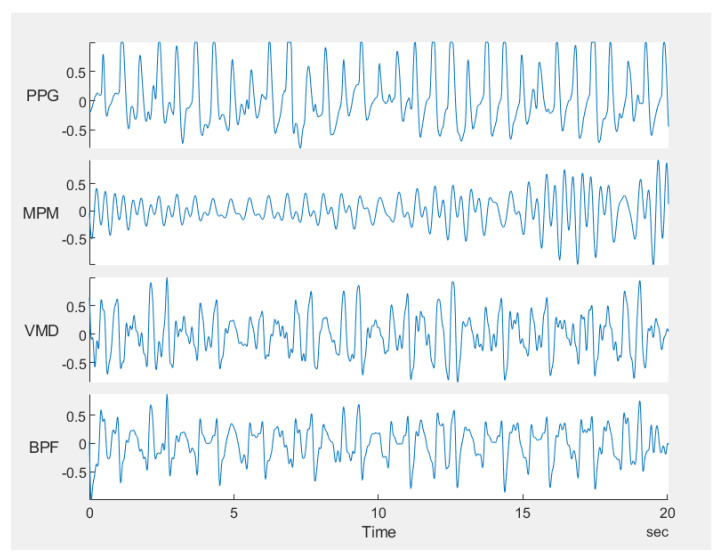
Comparison of heartbeat achieved by the UWB microwave sensor during abdominal crunches with different methods. PPG is the gold standard (first row). Rows 2–4 shows the heart pulsation signal obtained from the UWB sensor by applying MPM, VMD and BPF, respectively. For the sake of brevity, raw data have not been plotted.

**Figure 17 sensors-21-05735-f017:**
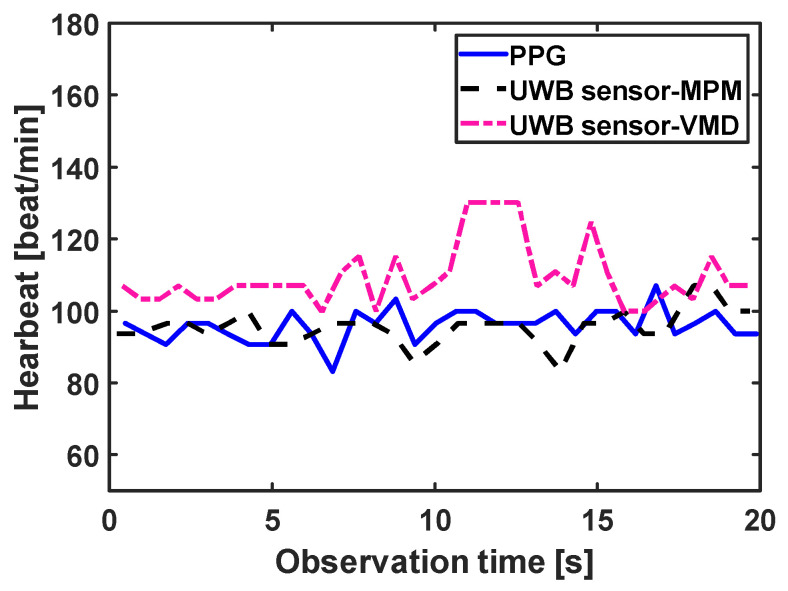
Calculated beat-to-beat heartbeat during abdominal crunch exercise by different signal processing methods. The signal was obtained by the UWB sensor. PPG is the gold standard.

## 5. Conclusions

The application of MPM for the extraction of vital signs from signals captured by microwave sensors was investigated. It was shown that the decomposition of the signal to its damping exponentials (natural poles) provides physical insights used to manipulate the natural poles regarding their frequencies and amplitudes and permits us to distinguish between noise, clutter, breathing/its harmonics, and heartbeat/its harmonics. Therefore, by keeping the desired natural poles and discarding the unwanted ones, one can easily reconstruct the wanted signal. In the context of heartbeat, it was shown that in addition to the average heartbeat, HRV is also traceable using MPM. The performance of MPM was compared with VMD as a new adaptive signal processing method and the conventional method of bandpass filtering. The algorithms were applied on a publicly available dataset captured by a CW sensor and the signals captured by a UWB sensor measured by authors. The results show that in less noisy scenarios such as directly sensing the carotid artery, all methods were quite similar. However, for noisy scenarios, e.g., when antennas were placed at a distance from the body, MPM showed the smallest error.

## Figures and Tables

**Figure 1 sensors-21-05735-f001:**
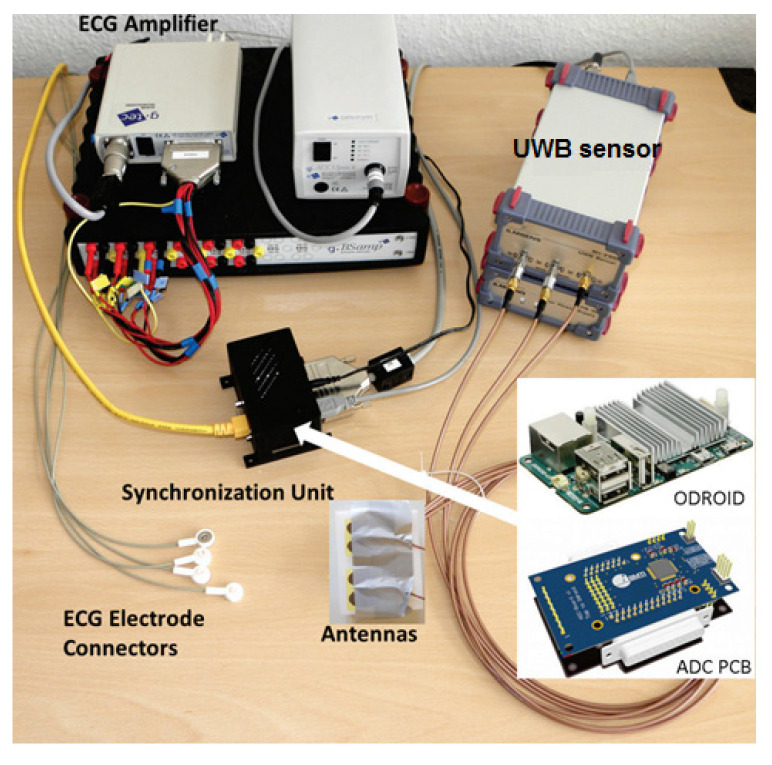
Measurement setup for the synchronized signal acquisition of ECG and UWB sensor.

**Figure 2 sensors-21-05735-f002:**
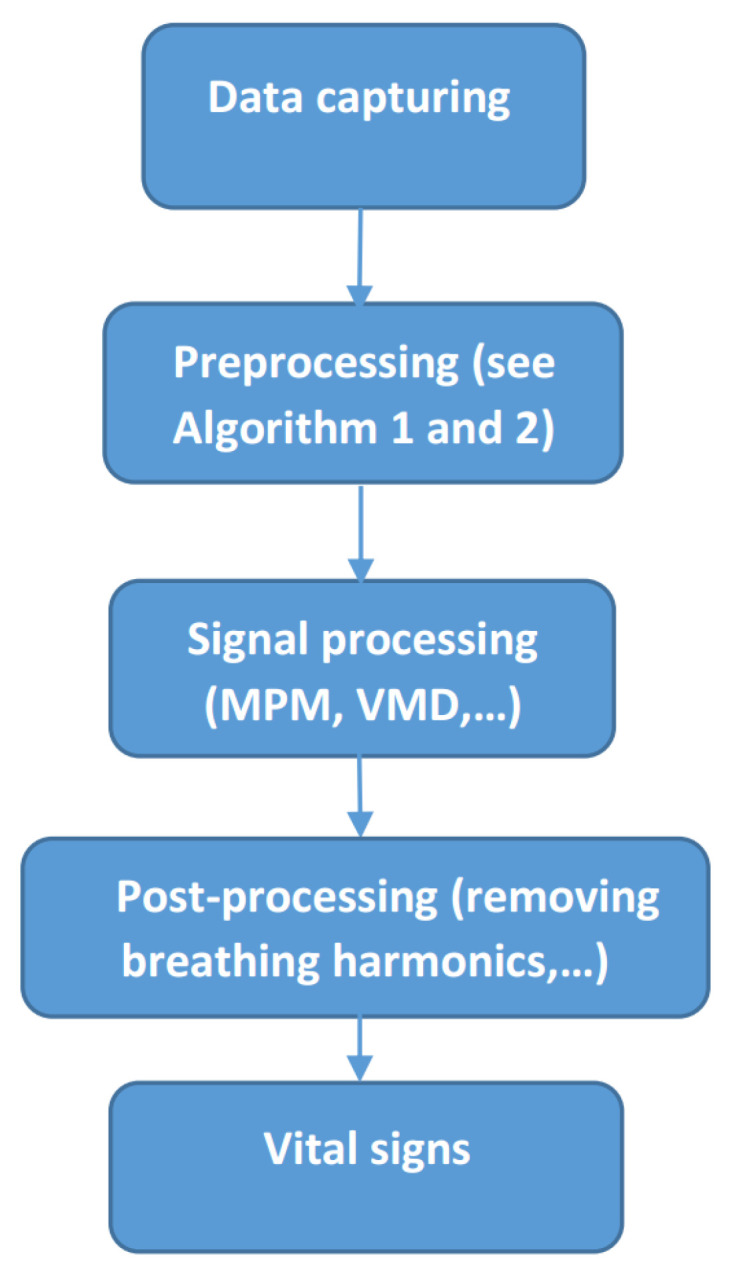
General flowchart of vital sign monitoring using microwave sensors.

**Figure 3 sensors-21-05735-f003:**
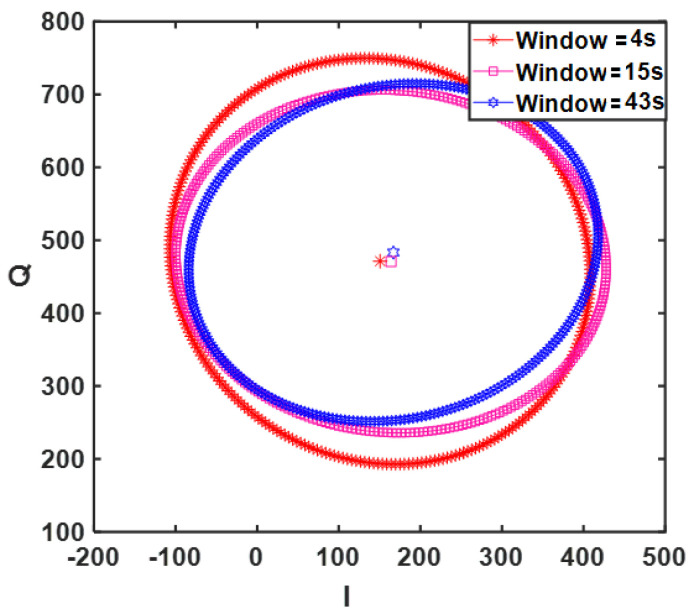
Ellipses fitted for different window sizes of CW sensor signal.

**Figure 5 sensors-21-05735-f005:**
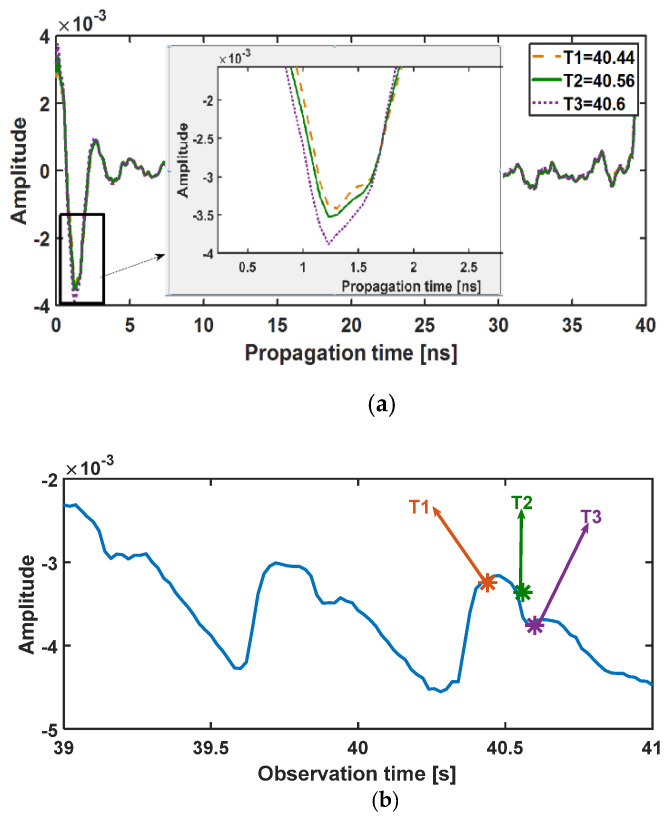
(**a**) Impulse response at three different observation times of T1, T2, and T3 and (**b**) the signal of range bin corresponding to 1.2 ns over the observation time. See also Figure 4b.

**Figure 6 sensors-21-05735-f006:**
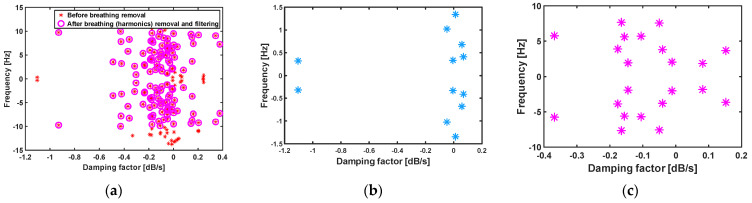
Natural poles of the first 16 s of signal captured by a CW sensor, GUARDIAN database, person 10, ID = 9-53-17; (**a**) total natural poles of signal, (**b**) natural poles of breathing and its harmonics, and (**c**) natural poles of heartbeat and its harmonics.

**Figure 7 sensors-21-05735-f007:**
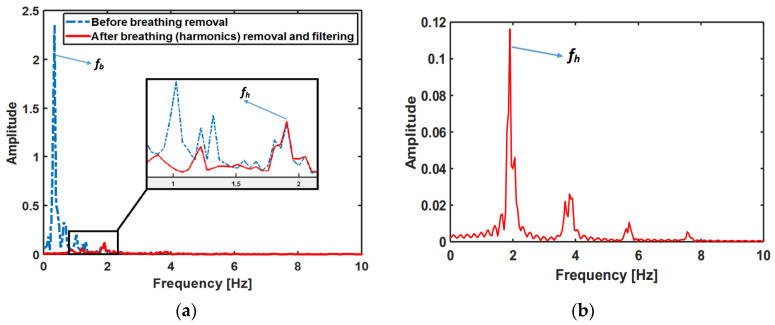
Absolute value of Fourier transform of the first 16 s of the signal captured by a CW sensor, GUARDIAN database, person 10, ID = 9-53-17; (**a**) before heartbeat and its harmonic selection and (**b**) after heartbeat and its harmonic selection.

## Data Availability

Data available on request.

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
