# Peer review of "Matrix Pencil Method for Vital Sign Detection from Signals Acquired by Microwave Sensors"

_sensors, 2021, doi:10.3390/s21175735_

Round 1

Reviewer 1 Report

The manuscript is well written and clearly explained.

Some minor comments were detected and given next.

Figure 2. Homogenize text font size. (Authors can include internal processes for pre- and post-processing).

Some figures describe processes, regardless of a general framework; algorithms could substitute some figures (Figs. 3 and 5).

In general, image captions are too synthetic; the authors can help the reader interpret images and overall research better.

In Sections 3.1 and 3.2, authors can avoid using acronyms in accordance with sections 3.3, 3.4 and 4.1 

In the singular decomposition shown in Eq. (12), V^H is not defined.

Some minor typographical details, Figure 8-Figure11 should be Figure 8-Figure 11 or Figures 8-11. 

Some references are incomplete; please revise all references, provides DOI if available.

Reviewer 2 Report

  1. Even though it has been mentioned several times in the article that when the vital sign is strong and pure, MPM, VDM, and BPF have almost similar performance, and in noisy cases MPM has better performance, these data are not shown in the experiment. Please provide the data regarding on your description.

  1. How could you get the Figure 9 (b)? By just removing the breath signal and its harmonics? What is the difference between Figure 9(b) with the curve “after breathing (harmonics) removal and filtering” in Figure 9(a)?

  1. You mentioned that “we experienced such an example when a sensor was placed at 1m distance from a subject during abdominal crunches…”, what type of a “sensor”? antenna or another sensor? It is recommended to depict this scenario by figure.

  1. Two parameters fb and fh are mentioned in line 106 and 109, respectively. It would be better to mark them in the corresponding figures.

  1. There is an extra period in line 25, and a period is missing in line 132.

  1. The beginning indentation is incorrect in line 144.
